# Autonomous Imagination: Closed-Loop Decomposition of Visual-to-Textual Conversion in Visual Reasoning for Multimodal Large Language Models

**Jingming Liu**[*]                                                                    *jml754457@gmail.com*
*State Key Laboratory of CAD&CG, Zhejiang University*

**Yumeng Li**[*]                                                                       *yumeng.li@zju.edu.cn*
*State Key Laboratory of CAD&CG, Zhejiang University*

**Boyuan Xiao**                                                                        *xiaoby202@gmail.com*
*State Key Laboratory of CAD&CG, Zhejiang University*

**Yichang Jian**                                                                    *mtdickens1998@gmail.com*
*State Key Laboratory of CAD&CG, Zhejiang University*

**Ziang Qin**                                                                       *qinziang19937@gmail.com*
*State Key Laboratory of CAD&CG, Zhejiang University*

**Tianjia Shao**                                                                        *tjshao@zju.edu.cn*
*State Key Laboratory of CAD&CG, Zhejiang University*

**Yao-Xiang Ding**[†]                                                                 *dingyx.gm@gmail.com*
*State Key Laboratory of CAD&CG, Zhejiang University*

**Kun Zhou**                                                                           *kunzhou@acm.org*
*State Key Laboratory of CAD&CG, Zhejiang University*

**Reviewed on OpenReview:** *https://openreview.net/forum?id=MI4yIBLprs*

## Abstract

Under pure textual modality, Large Language Models (LLMs) have demonstrated remarkable success in complex reasoning tasks by decomposing them into simpler sub-problems. However, Multimodal Large Language Models (MLLMs) still struggle with some seemingly straightforward visual tasks, such as counting and solving jigsaw puzzles. We argue that these tasks challenge the ability of *visual-to-textual conversion*, where MLLMs convert visual information perceived from the input scene, to textual information for further reasoning and generating the answer. If the complexity of the visual input is beyond the perceptual capability of the MLLMs, without decomposing this conversion process, simply scaling inference-time reasoning cannot solve the task because it repeatedly encounters the same perceptual bottleneck. We propose an approach, *autonomous imagination*, to enable MLLMs to iteratively modify visual inputs (e.g. isolating objects, rearranging puzzle pieces) into intermediate visual states, decomposing visual-to-textual conversion into closed-loop visual modification steps. We show that, without any retraining, MLLMs can now solve tasks initially beyond their perceptual capability, highlighting that closed-loop visual modification can be an effective way of decomposing the visual reasoning task into solvable substeps. Our code and data are released at `https://future-item.github.io/autoimagine-site/`.

---

[*]Equal contribution.
[†]Corresponding author.

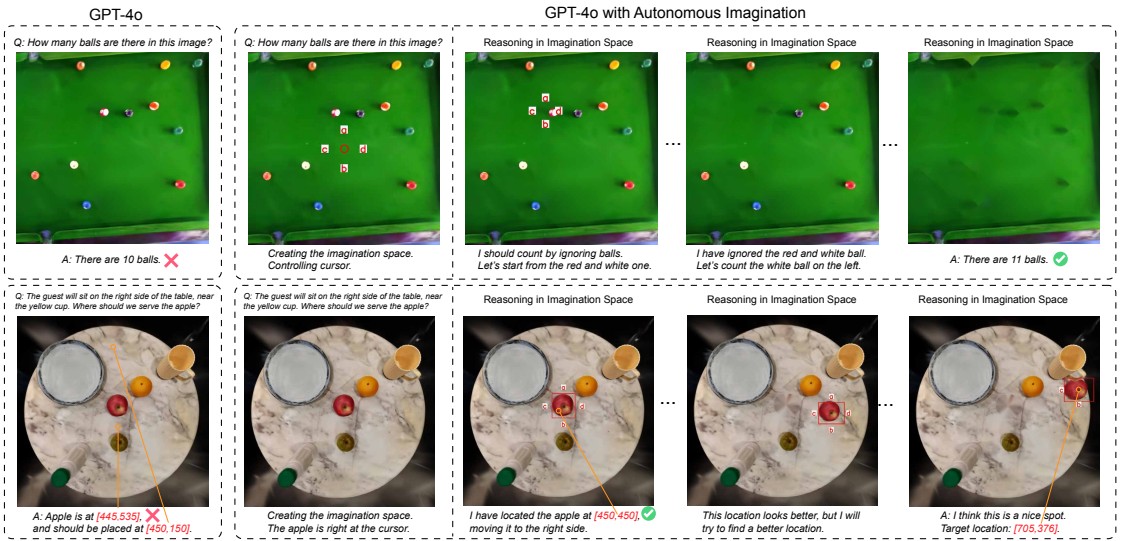

Figure 1: Our autonomous imagination approach empowers advanced MLLMs to engage in iterative imaginative reasoning, enabling them to address previously unsolvable tasks without additional training.

# 1 Introduction

Modern Large Language Models (LLMs) demonstrate exceptional reasoning capabilities through systematic task decomposition (Huang & Chang, 2023). By breaking complex problems into sequential subtasks, models achieve remarkable performance in textual reasoning domains (Yao et al., 2023; OpenAI, 2024b). More recently, Multimodal Large Language Models (MLLMs) (OpenAI, 2024a; Liu et al., 2023; Bai et al., 2023; Wang et al., 2023a) are gaining increasingly strong abilities in visual understanding. Inspired by work on visual prompting (Jiang et al., 2024; Lin et al., 2024; Hong et al., 2023; Zhang et al., 2023b;c; Mitra et al., 2023; Zheng et al., 2023; Zhang et al., 2024; Yu et al., 2024), recent methods enable MLLMs to perform visual reasoning by proactively adding visual prompts (such as bounding boxes) to the image, to reduce hallucination and handle challenging visual tasks without human intervention (Lai et al., 2023; Wu & Xie, 2023; Chen et al., 2023; Qi et al., 2024; Shao et al., 2024; Zhou et al., 2024b).

Despite these advancements, existing methods often struggle to make MLLMs natively handle intuitive visual reasoning tasks. When challenged with tasks such as jigsaw puzzle assembly (even when provided with explicit step-by-step instructions), or simply counting objects, state-of-the-art MLLMs consistently fail to reason through the entire process. A key observation is that, instead of challenging high-level reasoning abilities such as logical deduction, these tasks actually challenge the ability to perceive and understand complex visual scenes. Referring to the failure of existing MLLMs on them, we argue that ineffective task decomposition remains a fundamental cause. Reasoning in complex visual input requires the strong ability of *visual-to-textual conversion*, where MLLMs convert visual information perceived from the input scene to textual information for further reasoning and generating the answer. However, in current MLLM reasoning, the visual-to-textual conversion process remains an atomic operation rather than being divided into executable substeps. When the complexity of the visual scene is beyond the perceptual ability of MLLMs, without decomposing this conversion step, simply scaling inference-time reasoning cannot solve the task because it repeatedly encounters the same perceptual bottleneck.

We hypothesize that, similar to textual reasoning, where intermediate textual states are necessary, MLLMs need to generate intermediate images that gradually simplify the original scene or approach the target visual state. To test this, we propose enabling MLLMs to iteratively modify visual inputs (e.g., isolating objects, rearranging puzzle pieces) into intermediate visual states, decomposing visual-to-textual conversion into closed-loop visual modification steps, without any additional training of MLLMs. For simplicity, we refer to this approach as *autonomous imagination*. Technically, we implement the approach to support the

conversion from an unstructured input visual scene into a structured representation and the visual operators that efficiently modify the visual content into new states (e.g., through transformation or erasing). This allows MLLMs to generate successive visual states that progressively simplify the original visual scene or gradually approach the target visual state in an autonomous fashion.

We conduct experiments under visual reasoning tasks that challenge the perception and understanding abilities of MLLMs under complex visual scenes, including counting, jigsaw puzzle solving, object placement, and multi-object hallucination. The results show that previously challenging visual reasoning tasks, which are beyond the perceptual capability of the MLLMs, can be effectively tackled natively by them equipped with our approach. This serves as empirical validation that a major limitation in visual reasoning stems from insufficient support for visual task decomposition, which we hope will inspire future research.

## 2 Related Work

### 2.1 Reasoning in LLMs

Numerous studies explore reasoning paradigms in natural language using their in-context learning ability (Brown et al., 2020), showing that Large Language Models (LLMs) improve through reasoning-based outputs (Wei et al., 2022; Kojima et al., 2022). Subsequent work further improves reasoning through in-context learning by improving in-context sample selection (Rubin et al., 2021; Lu et al., 2022; Zhang et al., 2022; Fu et al., 2022; Wang et al., 2022; Li et al., 2022b). Recently, notable advancements in OpenAI's o1 (OpenAI, 2024b) have demonstrated that by scaling LLMs for inference-time reasoning before answering, LLMs can be greatly enhanced in reasoning. A key insight from these advances is that proper decomposition of the reasoning process into simpler subtasks is essential for successful reasoning (Wei et al., 2022).

### 2.2 Visual Reasoning in MLLMs

MLLMs have evolved in visual understanding ability, initially leveraging domain-specific expert models such as HuggingGPT (Shen et al., 2023), MM-REACT (Yang et al., 2023), and VisualChatGPT (Wu et al., 2023). The focus later shifted to training LLMs with an adapter for the other modal, such as LLaVA (Liu et al., 2023), BLIP-2 (Li et al., 2022a; 2023), and MoVA (Zong et al., 2024). Many recent MLLMs now exhibit native visual understanding through training in text-image pairs, with fine-tuning on question-answering tasks (Liu et al., 2023; Bai et al., 2023; Wang et al., 2023a; Zhu et al., 2023; Chen et al., 2022; Luo et al., 2023; Alayrac et al., 2022), including the state-of-the-art closed-source MLLM GPT-4o (OpenAI, 2024a).

In addition to reasoning within text modalities, substantial efforts are made in reasoning in visual tasks. Existing MLLMs often face limitations in direct visual perception and are prone to generating answers with hallucinations (Zhang et al., 2023d). Initial approaches used attention mechanisms to improve question-answering abilities (Zhou et al., 2021), and subsequent solutions include strengthening reasoning in the visual modality by training (Lin et al., 2023; Wang et al., 2023b), employing auxiliary knowledge (Mitra et al., 2023), and utilizing external visual perception modules Surís et al. (2023). In addition, researchers have proposed effective visual prompting methods to refine the focus of MLLMs (Jiang et al., 2024; Lin et al., 2024; Hong et al., 2023; Zhang et al., 2023b;c; Mitra et al., 2023; Zheng et al., 2023; Zhang et al., 2024; Yu et al., 2024). See Wu et al. (2024a) for the recent comprehensive survey. Attempts have been made to utilize visual prompts in the Chain-of-Thought (CoT) paradigm (Wei et al., 2022). Based on pioneer studies (Zhang et al., 2023d; Zheng et al., 2023; Zhang et al., 2023a; Peng et al., 2023), recent approaches build CoT by enabling MLLMs to add visual prompts autonomously either through direct model training (Lai et al., 2023; Wu & Xie, 2023; Chen et al., 2023; Qi et al., 2024; Shao et al., 2024) or by utilizing external visual processing models (Zhou et al., 2024b). By using visual prompts (e.g., bounding boxes) to highlight key information in images, the visual-to-textual conversion step becomes easier, leading to a performance boost. However, even though visual prompts simplify the perception of input images, this paradigm still restricts to do visual-to-textual conversion only in one step, which is after all visual prompts are added in the input. As our experiments demonstrate, even when visual prompting is applied, visual-to-textual conversion remains a bottleneck in common visual reasoning tasks that are intuitive to humans, highlighting the necessity of effective task decomposition.

We also discuss some recent work on action planning, which addresses a significantly different task from ours, while has also explored various imagination techniques, such as using video generation models to simulate control processes as references (Ajay et al., 2023; Du et al., 2023), employing image generation to visualize target goals (Zhou et al., 2024a), and improving an LLM's understanding of the current state, either through textual descriptions or visualizations (Liu et al., 2022; Wu et al., 2024b). However, these approaches are limited to closed-world settings where the state and action spaces are predefined. Although these techniques have shown promising results in closed environments, they are designed specifically for constrained settings and cannot be directly extended to open-world contexts, where the visual scene is primitive and unstructured.

Recently, advanced image editing models that accept input of natural languages are proposed (Huang et al., 2024; Wang et al., 2024). Notably, concurrent work has enables image editing models with strong reasoning abilities (Fang et al., 2025). These models have the potential to be utilized as the imagination tools in our approach, while are more capable to deal with visual generation and editing tasks by themselves.

The closest studies to ours are from the field of robotics, where recent methods utilize the ability of MLLMs to perform object manipulation (Kapelyukh et al., 2024; Ding et al., 2024). These methods also create a virtual space and use MLLMs as evaluators to judge whether the object's final state matches the instruction, in order to generate a final state where the object should be moved to. However, current approaches adopt random sampling to conduct exhaustive searches in the virtual environment, which is not feasible when the possible state space is large. We implement this method in the experiments to verify this argument.

## 3 Method

### 3.1 Problem Formulation

We challenge MLLMs with visual reasoning tasks that are intuitive for humans: *counting, jigsaw puzzle solving, object placement, and multi-object hallucination.* For object placement, to capture a holistic top-down view, we reconstruct the input scene using 3D Gaussian Splatting (Kerbl et al., 2023) and render it from a top-down perspective. We directly feed the unstructured raw input scene into the model: no semantic pre-processing (e.g., segmentation or grounding) is applied initially, aligning with real-world settings where visual inputs are provided directly. We evaluate whether MLLMs can natively solve these tasks by proactively modifying the visual state on their own, with minimal reliance on external semantic visual understanding abilities (e.g., we avoid using object detection models for counting, which would constitute "cheating").

### 3.2 Closed-Loop Reasoning Formulation

To enable existing MLLMs to proactively modify the visual scene during reasoning, we propose a plug-and-play virtual space. In this framework, the MLLM is treated as a pre-trained policy that executes a sequence of operations. We refer to this space as the *imagination space* for simplicity. The MLLM can move a cursor to focus on an object, perform an operation on it (e.g., ignore or transform it), and then go back to use the cursor again until convergence. The imagination space continuously updates the MLLM with the latest visual state by dynamically re-rendering the scene.

Note that our only external model dependency is Segment Anything 2 (SAM2) (Ravi et al., 2024), which is invoked solely to isolate a visual element after the MLLM selects it via the cursor. Since our focus is on testing visual perception rather than high-level logical reasoning, we directly provide the MLLM with task-specific reasoning plans (e.g., "count by ignoring balls") through system prompts. This mitigates occasional failures caused by the MLLM adopting bad reasoning strategies.

Specifically, we assume that the MLLM is given an input image $o$ and input text prompt token $r_0$, the aim of the reasoning is to obtain the prediction probability $P(y|o, r_0)$, where $y$ is the final output text token. In general, to utilize CoT, the reasoning problem of predicting $P(y|o, r_0)$ can be transformed into intermediate steps of predicting $P(z_t|o, z_{t-1}), t \in 1, 2, \ldots, T$. Each $z_t = \{r_t, c_t\}$ denotes the augmented textual prompts $r_t$ and visual information $c_t$ for deepening scene understanding and pushing forward reasoning. After the final reasoning step at $T$, the final output is set to $y = r_T$. The target transforms into predicting the following

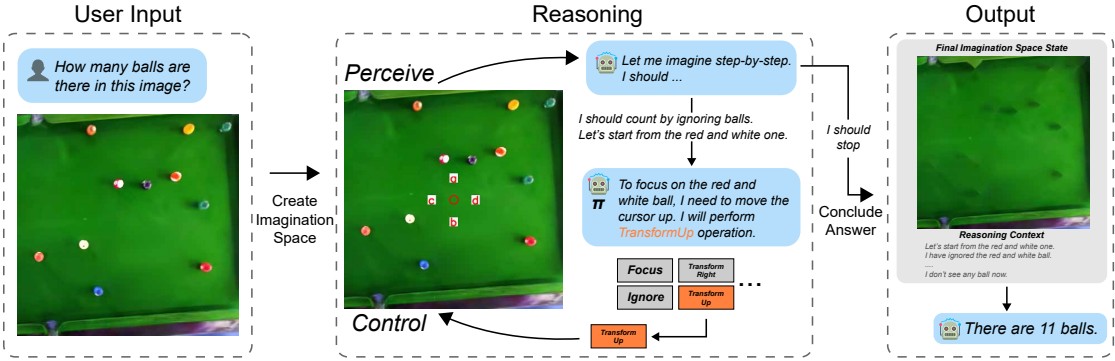

Figure 2: An overview of our autonomous imagination approach: The imagination space begins with an unstructured input scene and undergoes an iterative reasoning process. In each cycle, MLLMs first perceive the current state of the imagination space, select an operation to apply, and then reassess the updated imagination space. Upon completing this reasoning sequence, MLLMs generate an answer based on the cumulative context of the process and the final state of the imagination space.

joint probability:

$$P(z_{1:T}|o, r_0) = \prod_{t=1}^{T} P(z_t|o, z_{0:t-1}), \tag{1}$$

where $z_{t_1:t_2} = \{z_{t_1}, z_{t_1+1}, \ldots, z_{t_2}\}$ and $z_0 = r_0$. What is essential in CoT design is to make each step simple enough to match the reasoning capacity of MLLMs.

In imagination space, $\hat{o}$ are images rendered from the space for MLLMs to perceive. We also introduce a set of *operators* to modify the imagined scenes into new ones. Denote by $v_t = \{\hat{o}_t, r_{0:t}\}$ and $a_t$ the operator in step $t$. Our method transforms Eq. 1 into

$$P(\hat{o}_{1:T}, r_{1:T}, a_{1:T}|o, r_0) = \prod_{t=1}^{T} P(v_t, a_t|v_{t-1}), \tag{2}$$

where $v_0 = \{o, r_0\}$ and finally $y = r_T$. The one-step reasoning task $P(v_t, a_t|v_{t-1})$ is factorized into

$$P(v_t, a_t|v_{t-1}) = \pi(a_t|v_{t-1})\phi(\hat{o}_t|\hat{o}_{t-1}, a_t)\omega(r_t|\hat{o}_t, r_{0:t-1}). \tag{3}$$

Eq. 3 shows that one-step reasoning is factorized into a decision function $\pi(a_t|v_{t-1})$, a scene modification function $\phi(\hat{o}_t|\hat{o}_{t-1}, a_t)$, and a reasoning function $\omega(r_t|\hat{o}_t, r_{1:r-1})$. Given the current imagined scene $\hat{o}_{t-1}$ and the augmented text prompt token $r_{t-1}$, the decision function chooses a specific scene modification operator $a_t$. The scene modification function then utilizes $a_t$ to update $\hat{o}_{t-1}$ into $\hat{o}_t$. The reasoning function finally updates $r_{t-1}$ into $r_t$ based on $\hat{o}_t$. The decision and reasoning functions are purely based on the native reasoning ability of MLLMs. The scene modification function is implemented inside the imagination space.

Comparing Eq. 1 and Eq. 2, the imagination space transforms CoT reasoning into a closed-loop decision-making and reasoning process. During the reasoning process, the visual state is gradually simplified or approaching a target visual state: the reasoning step at step $t$ is only dependent on $\hat{o}_{t-1}, \hat{o}_t$ but independent of all previous visual states, leads to effective task decomposition.

## 3.3 Imagination Space

The imagination space is designed to render images for MLLMs to perceive and supports a minimal set of operators for MLLMs to call: focus, ignore, and transform. Focus isolates relevant content for further manipulation, ignore enables MLLMs to disregard extraneous information, and transform allows the repositioning of desired content. This compact set of operations enables MLLMs to reason and solve practical challenges by themselves, as demonstrated in our experiments. MLLMs select operators through natural language

output, which are then applied to the imagination space. The updated space is rendered and returned to MLLMs for the subsequent reasoning step.

We discuss the detailed implementation of the operators in the following sections. The operators are implemented separately for the 2D space (represented as images) and the 3D space (represented via 3D Gaussians). MLLMs are unaware of this implementation difference, as it interacts exclusively with rendered images as input and executes operators identically in both scenarios.

### 3.4 Focus Operator

The focus operator allows MLLMs to isolate and label target content from a scene, creating distinct elements for further transformation. Once MLLMs identify an object of interest using a virtual cursor (as described in Sec. 3.7), the focus operator segments this content for focused manipulation. In 2D space, this cursor directly conditions the Segment Anything Model (SAM) (Kirillov et al., 2023) to perform segmentation. MLLMs verify segmented output to ensure alignment with their intended focus. In the 3D Gaussian space, segmentation on demand poses unique challenges: Existing methods (Ye et al., 2024; Shen et al., 2024) are designed for unconditional segmentation, where all objects are segmented without the ability to specify conditions. However, our use case requires conditional segmentation, focusing on a specific object selected by an input condition, making these methods unsuitable.

To enable the focus operator in 3D Gaussian-based representations, we introduce a method to selectively segment an object from an unstructured 3D

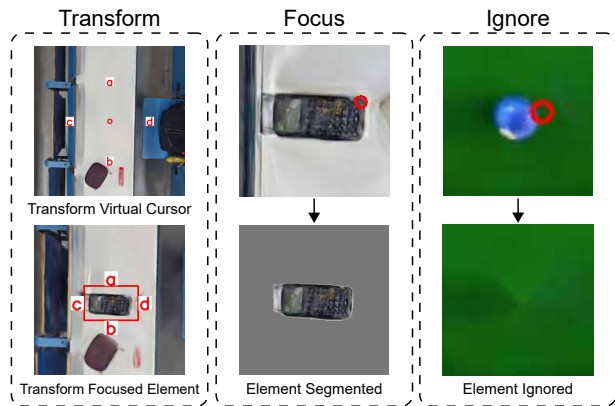

Figure 3: Illustrations of operations within our imagination space: transformations can be applied to focused elements (including the virtual cursor), focus operations allow segmentation of cursor-selected elements, and ignore operations make cursor-selected elements visually invisible.

Gaussian scene[1], which is illustrated in Alg. 1. Given the MLLM's initial selection block, we first generate a virtual camera trajectory that orbits around the targeted object, creating a video sequence. This sequence is fed into the SAM2 model (Ravi et al., 2024), which applies the selection cursor condition on all rendered frames. For each frame, rays are cast from pixels within the segmentation mask, and path tracing records the radiance contributions of intersecting Gaussians. This process is expressed as volumetric integration illustrated in Line 4-11. Gaussians who receive a contribution higher than a threshold will receive one vote.

We select the 3D Gaussian with votes exceeding 10% of the highest vote, forming a shell around the target object's radiance field. This shell, along with the internal radiance Gaussians, is segmented collectively to define the object's radiance structure. In both 2D and 3D spaces, the segmented object is placed in a separate layer, while the remaining content is retained in a base layer.

### 3.5 Ignore Operator

The ignore operation removes a focused object from the imagination space to prevent interference with the reasoning process. Since API access to state-of-the-art MLLMs does not support attention masks, removing the object leaves a hole in the base layer, potentially causing hallucinations. To address this, we simply inpaint the hole created by the removal of the object. For both 2D and 3D imagination spaces, we project the object mask onto the rendered image as the region to inpaint. We directly use the implementation provided by OpenCV (Itseez, 2015) by simply merging the nearby pixels. The inpainted image is then used

---

[1]We refer the detailed introduction of the 3D conditional segmentation algorithm to our open-released code.

---

**Algorithm 1:** 3D conditional segmentation

---

**Input** : SAM2 mask, 3D scene, thresholds $\epsilon_1$, $\epsilon_2$
**Output:** Set of marked Gaussians $\mathcal{G}_{\text{marked}}$

**1** Initialize vote counts: $\text{votes}[g_i] \leftarrow 0$ for each Gaussian $g_i$ in the scene
**2** **foreach** *pixel p in masked_pixels* **do**
**3**      $\text{ray} \leftarrow \text{cast\_ray}(p)$ ; $T \leftarrow 1$
**4**      **foreach** *Gaussian $g_i$ intersected by ray* **do**
**5**          $\alpha_i \leftarrow \text{get\_alpha}(g_i, \text{ray})$ ; $C_i \leftarrow T \cdot \alpha_i$
**6**          **if** $C_i > \epsilon_1$ **then**
**7**             $\text{votes}[g_i] \leftarrow \text{votes}[g_i] + 1$
**8**          **end**
**9**          $T \leftarrow T \cdot (1 - \alpha_i)$
**10**          **if** $T < \epsilon_2$ **then break**
**11**      **end**
**12** **end**
**13** $\text{max\_vote} \leftarrow$ maximum of $\text{votes}[g_i]$ over all $g_i$
**14** $\mathcal{G}_{\text{marked}} \leftarrow \emptyset$
**15** **foreach** *Gaussian $g_i$* **do**
**16**      **if** *$\text{votes}[g_i] \geq 0.1 \times \text{max\_vote}$* **then**
**17**          $\mathcal{G}_{\text{marked}} \leftarrow \mathcal{G}_{\text{marked}} \cup \{g_i\}$
**18**      **end**
**19** **end**

---

for further reasoning by the MLLMs, avoiding paying attention to the inpainted region as the content has been removed.

### 3.6 Transform Operator

Object transformation can occur in four directions: up, down, left, and right, each represented by an alphabet character $(a, b, c, d)$ to avoid interference with semantically meaningful words. In each step of the iterative imagination process, MLLMs select a direction for movement. Since MLLMs struggle with determining precise distances, we standardize movement units in screen space coordinates and gradually reduce step size throughout the process, where MLLMs only control the direction of movement.

### 3.7 Reasoning Process

The iterative reasoning process described in Sec. 3.2 is executed as follows: After initializing the imagination space, a virtual cursor is positioned at the center and automatically considered as focused. MLLMs can perform transform operations on the cursor to reposition it within the space. Following each operation, MLLMs conduct scene modification and receive an updated image of the newly rendered imagination space.

When MLLMs identify that the cursor has selected a visual element of interest, a focus operation is performed based on the location of the cursor. If MLLMs determine that this operation has correctly segmented the intended element, the focus is shifted to that object, enabling MLLMs to execute either a transformation or ignore operation on it. Once MLLMs have either repositioned or disregarded the object, the focus returns to the virtual cursor. This process iterates continuously as part of the reasoning workflow until all necessary reasoning steps are completed, culminating in a final textual response.

### 3.8 System Prompts

Our approach is training-free and utilizes the native reasoning ability of the MLLMs to autonomously utilize the visual imagination operators provided. We design a set of general system prompts across all tasks. The

reasoning instructions given to the MLLMs in our tasks are shown in Fig. 15. The prompts used as reminders for autonomous imagination operations are illustrated in Fig. 16.

## 4 Experiments

In the experiments, we challenge MLLMs with four tasks that are intuitive to humans and require strong visual understanding and reasoning abilities: counting, simple jigsaw puzzle solving, object placement, and multi-object hallucination. We target at verifying that compared to 1) simply scaling textual reasoning steps 2) conducting visual prompting, while still conducting visual-to-textual conversion within a single step, the closed-loop task decomposition paradigm is indeed more effective and necessary.

### 4.1 Benchmark and Evaluation Protocol

**Counting**: When faced with numerous densely packed objects, humans often rely on multi-step reasoning to reach an accurate count, as a single glance may not suffice. Existing MLLMs, however, struggle with counting, as their multistep reasoning is not yet effective for this task. We include this task to assess whether the model can leverage reasoning to offset its limited direct perception, mirroring human strategies.

For evaluation, we directly compare the predicted count of the model with the ground truth. In addition to reporting the **success rate**, we calculate the **mean** and **variance** of the counting errors to provide insight into the accuracy and consistency of each approach. We constructed 122 images with real objects with paired ground truth.

**Solving simple jigsaw puzzles**: Jigsaw puzzles are classic tests of visual perception and reasoning, commonly used to assess intelligence. In this task, we evaluate the ability of MLLMs to solve simple jigsaw puzzles, aligning their problem-solving performance with that of humans to assess visual reasoning capabilities. In the jigsaw puzzle solving task, MLLMs are tasked with identifying and placing missing pieces in their correct locations.

For evaluation, we use a digital jigsaw puzzle game with a magnetic mechanism, where pieces automatically snap into place when positioned close to their correct locations. Success is measured by the **completion rate**, defined as the percentage of pieces placed in their correct locations. For evaluation, we constructed 11 cases where four pieces are missing and 11 cases where six pieces are missing. The size of the puzzle ranges from $3 \times 5$ to $5 \times 8$.

**Object placement**: Previous methods have shown progress in enabling MLLMs to understand and describe static scenes, such as identifying object locations (e.g., "Where is the cup?"). However, for practical use, MLLMs must also interpret dynamic instructions that convey intent, such as "Where should the cup be placed?" In the object placement task, MLLMs are required to identify both the current and target locations of objects based on abstract instructions (e.g., "Prepare two cups for guests in the living room"). Given the input scene of 3D Gaussians, MLLMs must determine the original locations of objects and their intended locations based on a provided prompt.

For evaluation, we first measure the **locating success rate**, which reflects the model's ability to accurately identify the initial location of the correct object. If the model fails to do this, the placement task is automatically marked as a failure. We then measure the **placement success rate** by checking whether the predicted final location falls into the marked ground-truth region. For both success rates, when multiple correct solutions exist, we provide multiple ground truth regions and selecting any valid region is considered correct. We captured a total of 17 scenes, including 271 user prompts and paired ground truth for evaluation.

**Multi-object hallucination**: Current MLLMs can suffer from hallucination by perceiving or generating non-existing objects in the input scene. The complexity of the input scene is beyond the limitation in perceptual capability of MLLMs, which is a plausible reason for hallucination. Our approach can serve as a promising way to address this challenge, especially for the scenes where multiple objects exist. By the closed-loop iterative process of autonomous imagination, MLLMs can deal with objects one by one, as in our counting benchmark, instead of handling them all together, effectively reducing the possibility of hallucination. We conducted additional experiments on the recently proposed multi-object hallucination

Table 1: Quantitative comparison results under counting, jigsaw puzzle solving, and object placement. See Sec. 4.1 for details of the metrics. We consider two variants of cursor-only in counting, leading to two sets of results, and GPT-4o Sampling cannot be applied to counting and locating as illustrated, please see Sec. 4.3, 4.2, and Sec. 4.5 for details. *In object placement, cursor-only functions the same as our method in locating, so their results are identical. Furthermore, NA* indicates that under these tasks, Molmo constantly fails to output reasonable coordinates or refuses to output.

| Settings | | GPT-4o | o1 | Molmo | VCoT | Ours (cursor-only) | GPT-4o Sampling | **GPT4o+Ours** |
|---|---|---|---|---|---|---|---|---|
| Counting | Success Rate | 39.8% | 50.8% | **86.2%** | 15.5% | 39.0%/0% | - | 85.3% |
| | Mean Error | 0.73 | 0.74 | 0.34 | 3.11 | 1.12/*not calculable* | - | **0.19** |
| | Variance | 0.62 | 0.90 | 2.57 | 5.51 | 1.22/*not calculable* | - | **0.22** |
| Simple Jigsaw Puzzle | 4-Piece Missing | 29.5% | 31.8% | NA* | 9.1% | 27.3% | 43.2% | **68.2%** |
| | 6-Piece Missing | 9.1% | 27.3% | NA* | 3.3% | 24.2% | 30.3% | **51.5%** |
| Object Placement | Locating | 10.9% | 17.3% | NA* | 10.4% | **69.4%**\* | - | **69.4%** |
| | Placement | 3.6% | 8.5% | NA* | 1.5% | 27.8% | 17.3% | **37.3%** |

benchmark ROPE (Chen et al., 2024a) to validate this argument. We focus on the most challenging subset in the benchmark where the state-of-the-art model on the official leaderboard, GPT-4o, exhibits the poorest performance, which is answering about single object given multiobject image, heterogeneous object types, on unseen data split. We refer to the detailed benchmark setting from the original paper.

## 4.2 Baselines

We evaluate our approach against state-of-the-art MLLM, namely **GPT-4o** (OpenAI, 2024a) (our approach is also utilized to enhance GPT-4o in all experiments). We provide a clearly defined 2D coordinate system to GPT-4o so that output coordinates are generated without any ambiguity. We also adopt the recently proposed visual reasoning method **VCoT** (Shao et al., 2024) using their open-released pre-trained model. We developed a baseline **GPT-4o Sampling** inspired by the sample-then-evaluate imagination paradigm used in robotics (Kapelyukh et al., 2024; Ding et al., 2024). Since their original methods are designed for robotic manipulation, we reimplement them under our method. Note that this sampling strategy can only be applied to some of the tasks in our evaluation since sampling in large solution spaces is heavily intractable. Furthermore, to demonstrate the necessity of altering the visual scene for task decomposition beyond adding visual prompts, we crafted a reasoning baseline named **cursor-only** that utilizes our imagination space, but restricts operations solely to transform operations of the virtual cursor. Note that this baseline serves as an ablation of our method, which has the same closed-loop control design except that scene modification is disabled and only virtual cursor moving is enabled. Furthermore, we also adopt two recently proposed advanced MLLMs, namely **OpenAI's o1** (OpenAI, 2024b) and **Molmo** (Deitke et al., 2024). o1 is well known to have specifically trained reasoning abilities under pure textual modality. We utilize it to justify whether simply scaling text-time textual reasoning could solve the tasks. On the other hand, Molmo is trained to have ability to utilize visual markers as cues in visual modality for reasoning. We utilize it to justify the necessity of task decomposition instead of conducting visual prompting with visual markers.

## 4.3 Counting

The results are shown in Tab. 1. Our approach significantly improves the performance of original GPT-4o, achieving a higher success rate and lower errors when mistakes occur. It is worth noting that Molmo is known to be specifically trained to conduct counting with its native "pointing" ability. Our approach enables GPT-4o to perform on par with it. Note that the sampling method is not directly applicable to the counting task due to intractability, so we omit its comparisons. VCoT only supports drawing one bounding box to highlight a piece of visual evidence, hence is unsuitable for more complex reasoning tasks involving multiple objects, such as counting. We also compare with our cursor-only baseline in counting, observing that it performs similarly to GPT-4o. Since having a cursor moving around might be ineffective for counting, we adopt a cursor-only variant, where the MLLM can draw a bounding box on the cursor location to highlight the object during counting. Although technically able to draw boxes on every ball to perform counting

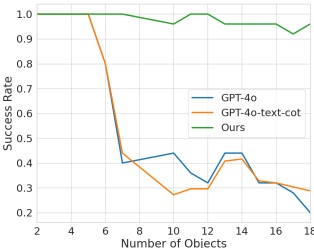 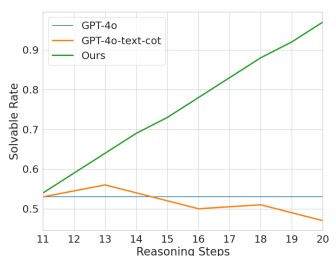 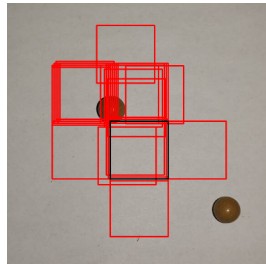

(a) Success Rate vs. Number of Objects     (b) Solvable Rate vs. Reasoning Steps     (c) Counting by drawing bounding boxes only

Figure 4: (a)(b) show that as counting task difficulty increases linearly, scaling inference-time textual reasoning (implemented as GPT-4o-text-cot, See Sec. 4.3) fails—and even performs worse than vanilla GPT-4o—as complexity exceeds perception limits. In contrast, our methods remain unaffected by these limits, achieving correct counting even as difficulty rises. Additionally, adding visual prompts can introduce noise, causing MLLMs to enter hallucination loops; for instance, in (c), the model incorrectly concluded there were 196 balls when only two were present. The qualitative results of counting are provided in the appendix.

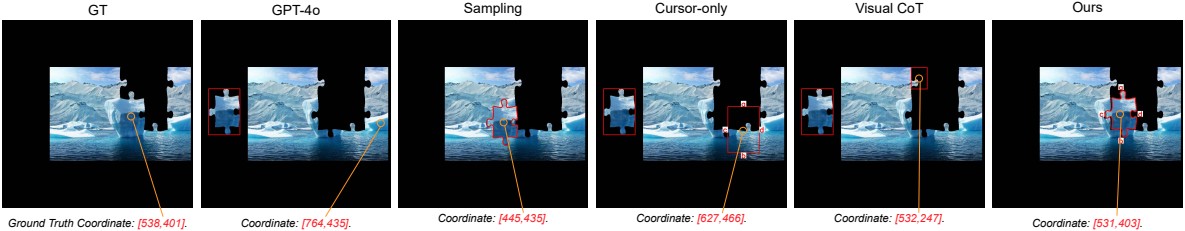

Figure 5: Qualitative comparison on simple jigsaw puzzle solving, we use a black background to make jigsaw pieces more visible to MLLMs. We illustrate the final visual state achieved by different methods after completing their reasoning processes and producing a solution. For clarity, the actual location of each coordinate in the image is highlighted with an orange line and circle.

accurately, the model quickly enters a negative feedback loop: hallucinations lead it to draw more markers, which introduce additional noise and exacerbate the hallucinations. This results in a success rate of 0% and makes mean error and variance not calculable, as the model cannot stop counting, further highlighting the importance of modifying the visual scene.

To further verify the essentialness of task decomposition in visual modality, we conduct additional analysis under the real object data by increasing the counting difficulty by adding more objects with partial data. The purpose was to evaluate the impact of difficulty and reasoning steps on performance. To achieve this, we selected a subset of cases from the whole data and supplemented them with simpler cases involving fewer objects. We then re-conducted the experiments on this partial dataset using GPT-4o, alongside a newly introduced textual CoT reasoning baseline. In the first experiment (Fig. 4a), we show the success rate of different methods when faced with varying numbers of objects. In the second experiment (Fig. 4b), we test how different methods improve when given different limits on the number of reasoning steps. The result is shown as the solvable rate, which indicates the percentage of cases the method can solve under the current budget. When imposing such a limit, our model is restricted to performing no more than the specified number of steps, with each step defined as a combination of the focus operation and the subsequent operation performed after focusing. For the textual CoT baseline, we explicitly instructed the model on the maximum number of reasoning steps it was allowed to use. The results are shown in Fig. 4a with an additional baseline **GPT-4o-text-cot**, indicating the pure text-based CoT. This reveals perceptual limitations in GPT-4o that cannot be overcome by simply increasing the textual CoT steps, while our method remains unaffected. Fig. 4b shows the percentage of cases solved as reasoning steps increase. For reference, we include the one-step results from GPT-4o depicted as a flat line. Our method performs consistently better as reasoning steps

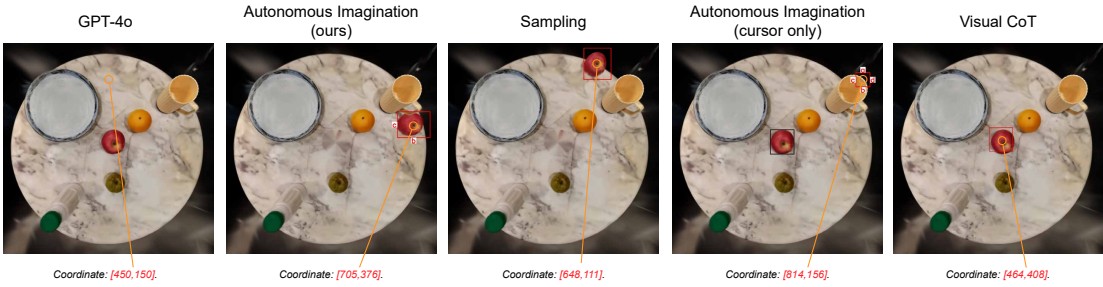

Figure 6: Qualitative comparison of apple placement based on user prompts. Coordinate locations are highlighted on the image with orange lines and circles for improved visualization. Our method aligns more closely with the user's requirements, placing the apple further to the right side of the table.

grow, whereas GPT-4o-text-cot fails to do this and may even introduce noise, resulting in poorer performance than one-step GPT-4o.

## 4.4 Simple Jigsaw Puzzle Solving

In the jigsaw puzzle solving task, the rotations of pieces are not considered. As multiple trials are allowed, we restrict the total number of attempts to 20 and report the finish rate, defined as the proportion of missing pieces successfully placed. The sampling method involves comparing pairs of target locations and iteratively selecting the better option until a single target location remains. We guarantee that the correct answer exists within the sampling process. In our method, once a jigsaw piece is moved to the imagined target location, it is considered to be placed. GPT-4o and o1 select the target location by outputting coordinates, while VCoT and Molmo determine the target location by drawing a block or marker, leveraging their specialized training for such tasks.

As shown in Tab. 1 and Fig. 5, our method consistently outperforms all baselines. It achieves a higher locating accuracy, which contributes to a higher success rate with the same number of attempts. Since this visual reasoning task requires the model to choose the correct target location rather than simply locating visible elements, VCoT is ineffective at reasoning. The sampling method performs reasonably well in this task, since we set a large sampling budget to ensure the correct answers can be sampled as the candidate answers. However, despite its extensive reasoning budget, it still does not match the performance of our method. This highlights the effectiveness and robustness of establishing an autonomous perception-control loop. We also demonstrate that the cursor-only baseline performs similarly to primitive GPT-4o in the four-piece missing scenario. However, it performs significantly better under the more challenging six-piece missing scenario, further confirming the effectiveness of our closed-loop control approach.

## 4.5 Object Placement

Similar to the approach used in solving simple jigsaw puzzles, the methods compared include the following: GPT-4o and o1, which output the target location by specifying coordinates; the sampling method, which iteratively compares sampled results until a single option remains; VCoT and Molmo, which determines placement by drawing blocks or markers; and our method, which outputs the final placement after completing a transformation operation. In line with how related work in robotics handles sampling-based methods (Kapelyukh et al., 2024; Ding et al., 2024), the sampling baseline restricts its search to a subregion of the space, filtering out certain incorrect answers. For example, regions occupied by existing objects are excluded from consideration. This principle is also incorporated into the transformation operation in our method. Specifically, a focus operation is first applied to identify the platform on which the target object should reside, creating a region mask. The transformation operation then ensures that no movement is suggested if it would lead the object outside the defined mask. This approach improves efficiency and ensures

Table 2: Results under multi-object hallucination benchmark. Except for o1, Molmo, and our method, the results of other baselines are cited from the official leaderboard in `https://multi-object-hallucination.github.io/`. For each base model family, we take the one with the highest performance. Mechanistically grounded MLLMs (marked with *) take visual prompts by dedicated pointer tokens. Please refer to the original paper for more details.

| Models | Acc. | Models | Acc. | Models | Acc. | Models | Acc. |
|---|---|---|---|---|---|---|---|
| LLaVA-34B | 30.81% | CogVLM-C | 13.50% | Qwen VL-C | 15.37% | Molmo | 28.90% |
| IDEFICS | 6.50% | GLaMM* | 52.28% | MiniCPM-V | 14.39% | o1 | 60.20% |
| Yi-VL-34B | 0.41% | GroundHOG* | 38.13% | GPT-4o | 53.74% | GPT-4o+Ours | **65.00%** |

Table 3: Results under more existing benchmarks.

| Models | CLEVR (Counting) | | | Where2Place (Object Placement) |
|---|---|---|---|---|
| | Success Rate | Mean Error | Variance | Accuracy |
| GPT-4o | 57.3% | 0.52 | **0.45** | 29.1% |
| GPT-4o + ours | **74.7%** | **0.42** | 0.74 | **37.0%** |

a fair comparison between the sampling baseline and our method by reducing unnecessary exploration of invalid regions.

As shown in Tab. 1, our method consistently outperforms the baseline methods in both object locating and placement performance. It is important to note that our cursor-only ablation baseline is identical to our full method when locating an object. Therefore, we assign them with the same locating result. Additionally, the sampling method is intractable for locating. Thus, we directly utilize the locating results of our method for placement. Methods lacking advanced visual reasoning capabilities perform poorly on the placement task, as successful placement requires inferencing about the correct spatial position for that element beyond visual recognition. The cursor-only baseline, which moves the visual marker instead of the scene modification, does not perform as effectively as our complete method. This difference underscores the necessity for substantial modification in visual task decomposition. Although the sampling baseline receives substantial visual information, it still underperforms relative to our method and even falls short of the cursor-only baseline. In contrast, in simple jigsaw puzzle solving, the sampling method achieves a relatively high success rate compared to the cursor-only baseline. This contrast highlights the importance of a structured reasoning pathway, particularly in complex, open-world scenarios where the abundant visual information could overwhelm the MLLM, preventing it from identifying the correct answer despite its presence in the sample. By following the closed-loop control reasoning process, MLLMs can progressively approach the correct answer without requiring an exhaustive number of samples, resulting in greater efficiency and improved accuracy.

### 4.6 Multi-Object Hallucination Benchmark

We evaluated our method using the recently proposed ROPE benchmark for Multi-Object Hallucination (Chen et al., 2024a), focusing on the most challenging subset where GPT-4o exhibits the poorest performance, which is answering about a single object given multi-object image, heterogeneous object types, on unseen data split. Our method uses GPT-4o as the base model that equips the imagination space. Please refer to the original paper for more details. Given that this benchmark emphasizes the identification of abstract objects that are not suitable for segmentation, we turn the focus operation into drawing a large rectangular region by MLLMs natively, through selecting top-left and bottom-right corners. The benchmark indicates that MLLMs are prone to hallucinations when confronted with multiple objects that introduce additional visual distractions. We demonstrate that using our method, MLLMs can autonomously focus on the important region despite these distractions, leading to more accurate responses due to the elimination of irrelevant visual information.

Table 4: Results for smaller MLLMs under the object placement task.

| | Qwen2.5-VL-7B | | Qwen2.5-VL-32B | | InternVL2_5-8B | | InternVL2_5-26B | |
| | Base Model | Ours | Base Model | Ours | Base Model | Ours | Base Model | Ours |
|---|---|---|---|---|---|---|---|---|
| Locating | 7.2% | **19.3%** | 2.9% | **16.8%** | 1.1% | **11.6%** | 1.5% | **16.7%** |
| Placement | 1.1% | **2.2%** | 0.85% | **4.2%** | 0% | **3.6%** | 0.7% | **5.1%** |

Table 5: Comparison between RobotPoint (fine-tuned to solve Where2Place tasks) and GPT-4o enhanced with our approach.

| Models | Counting | | | Simple Jigsaw Puzzle | | Object Placement (Ours) | | Object Placement (Where2Place) |
| | Success Rate | Mean Error | Variance | 4-Piece Missing | 6-Piece Missing | Locating | Placing | Accuracy |
|---|---|---|---|---|---|---|---|---|
| RobotPoint | 25.0% | 2.04 | 3.92 | 6.8% | 6.1% | 1.0% | 0.03% | **46.8%** |
| GPT-4o | 39.8% | 0.73 | 0.62 | 29.5% | 9.1% | 10.9% | 3.6% | 29.1% |
| GPT-4o + ours | **85.3%** | **0.19** | **0.22** | **68.2%** | **51.5%** | **69.4%** | **37.3%** | 37.0% |

## 4.7 Discussions

**Results on more existing benchmarks**. To verify the robustness of our approach, we further conduct experiments under two more existing benchmarks. One is CLEVR (Johnson et al., 2017) for the counting task. In each image from CLEVR, a scene containing various number of geometric objects is presented. We randomly sample from the official test set to ensure balance over the numbers of objects in the images, which includes 178 images. Another is Where2Place (Yuan et al., 2024), which includes 100 real-world scenes from homes and offices in the wild for the object placement task. The results are shown in Tab. 3, verifying that our approach consistently improves the reasoning ability of the base MLLM.

**Applying on smaller-sized MLLMs**. Our approach requires the ability of the base MLLMs to autonomously utilize the visual imagination operators provided. This is natively satisfied by larger state-of-the-art MLLMs, such as GPT-4o, while maybe challenging for smaller-sized MLLMs without specific training. We verify this in the counting and jigsaw puzzle solving tasks, where the smaller-sized MLLMs usually have the difficulty in realizing "when to stop": Common issues include incorrectly deciding to stop counting when there still exist remaining objects, or being hard to judge whether the jigsaw piece has been put at the right position. This shows the essentialness of studying how the autonomous imagination ability can be trained, which is left as future work. On the other hand, we discover that our training-free strategy can indeed be utilized to enhance smaller-sized MLLMs (Qwen2.5 (Bai et al., 2025) and InternVL2.5 (Chen et al., 2024b)) under the object placement task, which is shown in Tab. 4, verifying the potential of our approach for smaller-sized MLLMs.

**Comparing with in-domain training**. It is also meaningful to compare our approach with in-domain trained models. In Tab. 1, we have shown that Molmo achieves strong performance in the counting task, with the specifically trained ability to utilize visual cues in counting, while performing not ideally under other tasks. We further test the RobotPoint model (Yuan et al., 2024), trained specifically for predicting image keypoint affordances given language instructions, which is especially suitable to handle the Where2Place benchmark. We compare its performance under Where2Place, as well as our counting, jigsaw puzzle solving, and object placement tasks, as shown in Tab. 5. Similar to Molmo, RobotPoint shows strong performance under Where2Place that it is trained to handle, while fails to generalize its ability in other tasks. This suggests that in-domain training is more suitable to address specific tasks, while improving test-time reasoning can be an easier way to improve general reasoning ability across different tasks. Moreover, as discussed in the analysis of smaller-sized MLLMs, conducting training on the general reasoning ability instead of domain-specific task solving is also promising.

**Concurrent work on multimodal CoT reasoning**. Concurrent to our work, there is a surge of research for multimodal CoT reasoning. A representative is OpenAI's o3 model (OpenAI, 2025a), which is natively trained multimodal reasoning ability for "thinking with images" (OpenAI, 2025b), which utilizes visual processing tools to modify images (e.g. crop, zoom, rotate), hence realize more thorough analysis of the inputs during CoT reasoning. This paradigm is also studied by other works via visual test-time scaling (Luo et al., 2025; Wu et al., 2025), supervised fine-tuning (Li et al., 2025; Fu et al., 2025), and reinforcement

Table 6: Comparison between OpenAI's o1 and o3 models under our benchmarks.

| Models | Counting | | | Simple Jigsaw Puzzle | | Object Placement | |
|---|---|---|---|---|---|---|---|
| | Success Rate | Mean Error | Variance | 4-Piece Missing | 6-Piece Missing | Locating | Placing |
| o1 | 50.8% | 0.74 | 0.90 | 31.8% | 27.3% | 17.3% | 8.5% |
| o3 w/o tool | 50.0% | 0.71 | 0.78 | 61.4% | 66.7% | 9.5% | 4.4% |
| o3 | 76.2% | 0.36 | 0.69 | 78.6% | 57.1% | 22.9% | 6.2% |

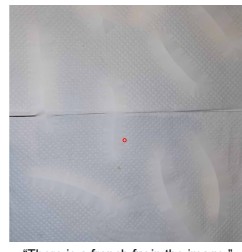 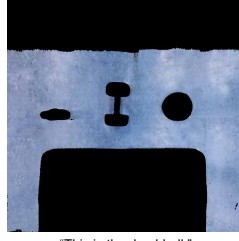 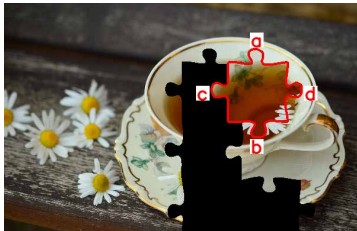

"There is a french fry in the image."     "This is the dumbbell."     "I should place this piece at this spot."

Figure 7: Failure cases in our method caused by hallucination, including instances where MLLMs mistakenly perceive a fry that does not exist, misinterpret the wrongly focused floor as a dumbbell, and incorrectly believe that a puzzle piece has been placed in the correct spot.

learning (Duan et al., 2025; Xu et al., 2025). Tab. 6 illustrates the comparison among OpenAI's o1 and o3 models in our proposed benchmarks, with an additional baseline "o3 w/o tool" in which we explicitly prompt o3 to disable the tool use ability. The results are consistent with our work: Multimodal CoT usually improves over purely textual reasoning. Compared with these concurrent works, our work uniquely proposes the notion of "visual-to-textual" conversion and explicitly differentiates between visual prompting and visual imagination. This can lead to better understanding of a fundamental research question: *In what kinds of tasks, the ability to think with images is necessary?* Furthermore, our plug-and-play autonomous imagination approach can be of independent interest to inspire more advanced multimodal CoT reasoning approaches.

**Failure cases**. In Fig. 7, we illustrate failure cases of our approach, which are majorly resulted from the hallucination of MLLMs.

**Time cost**. The time cost of our approach depends mainly on three factors: the running time of a single call of imagination operators, a single call of MLLM reasoning, and the number of MLLM calls. The calls of imagination operators are quite efficient. For example, each call of the SAM model takes 220ms on NVIDIA RTX 4090 on average, which is very efficient compared with MLLM reasoning. Considering the number of calls for MLLM reasoning, the average numbers of MLLM API call for our approach are: counting=117 (for all objects), jigsaw puzzle=20 (for one piece), object placement=15 (for one object), multi-object hallucination=26 (for one object). As shown by the comparisons on GPT-4o Sampling baseline in Tab. 1, our approach is more efficient than the closest previous studies to ours: sampling-based imagination strategies (Kapelyukh et al., 2024; Ding et al., 2024), by enabling to solve broader tasks with vast solution spaces.

## 5   Conclusion

In this work, we target at tackling visual reasoning problems using MLLMs where textual-to-visual conversion is the major bottleneck. We propose the autonomous imagination approach, which employs plug-and-play imagination space and operator design, enabling MLLMs to modify visual content autonomously, leading to closed-loop decomposition of this conversion process. We conduct experiments under visual reasoning tasks that are beyond the perceptual capabilities of MLLMs, while remained straightforward to humans. The results show that these tasks can be effectively tackled natively by MLLMs equipped with our approach.

**Limitations and future work**. Our work is a proof of concept: While closed-loop task decomposition overcomes the visual-to-textual conversion bottleneck in our demonstrated tasks, resource constraints pre-

vented us from training a model with native closed-loop reasoning and visual modification capabilities. This also prevents us from validating our approach under large, general-purpose visual benchmarks. We hope that our work inspires future efforts to close this critical gap. Furthermore, even though we test the robustness of our approach in counting and jigsaw puzzle sovling through different levels of difficulty (in terms of numbers of objects/pieces), for object placement and multi-object hallucination tasks, properly measuring the task difficulty is itself a challenging problem. These tasks involve real-world scenes with complicated spatial and semantical relationships among the objects, which may not be easily captured by a single measurement of hardness. We treat this as an important future research problem. Finally, our current approach limits to utilize chain reasoning with fixed plans, which is not optimal in terms of time cost. Future work can improve the efficiency of reasoning with more advanced searching and planning strategies.

## Acknowledgement

This work was supported by National Natural Science Foundation of China (U23A20311,62206245). We would like to thank the reviewers for their insightful and constructive suggestions.

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

Table 7: Quantitative results on counting using Cog-CoM (Qi et al., 2024), by directly asking it to output the number of objects, and by prompting it to use the trained GROUNDING ability, which counts for the object. We find that though explicitly trained to perform GROUNDING which counts the specific object, it somehow performs worse in our evaluation.

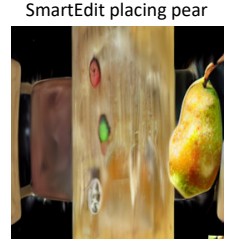 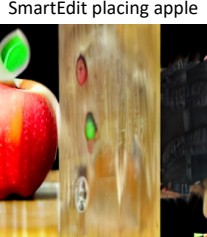

SmartEdit placing pear    SmartEdit placing apple

|  | Settings | CogCoM | CogCoM-GROUNDING |
|---|---|---|---|
| | Success Rate | 30.3% | 14.8% |
| Counting | Mean Error | 1.26 | 3.77 |
| | Variance | 1.39 | 15.47 |

Figure 8: We show that existing image editing models lack the capability to effectively comprehend and execute complex commands, such as placing the apple/pear at the left/right side on the table.

**Input**

**Output**

Identify the empty region in the image. The landmark information does not provide sufficient details to identify a specific area within the given description "sa_1679". So ultimately, the conclusive answer to the question in discussion is doesn't match my current knowledge.

Figure 9: Qualitative results on using CogCoM (Qi et al., 2024) for jigsaw puzzle solving show that it generates unrelated text when asked where the highlighted puzzle piece should be placed.

## A    Potential Broader Impact

Our work is largely foundational and aims to improve the reasoning capabilities and reliability of multimodal language models. Reducing hallucinations in complex visual scenes could enhance safety and robustness in real-world applications. On the other hand, our approach could increase the computational cost of MLLMs, leading to energy inefficiency and environmental impact. Reproducing our work also requires access to existing MLLMs and visual tools, leading to additional complexity and costs. We hope that our efforts on open-release all our code and data could reduce this burden.

## B    More Experiments

**Imagine with Image Editing Models.** We experimented with utilizing image editing models that accept input of natural languages, such as the state-of-the-art model SmartEdit (Huang et al., 2024). Ideally, such models would provide effective guidance by generating the target goal based on descriptions in natural language. However, as shown in Fig. 8, we find that the image editing model is not ideal when it comes to following complex instructions. This underscores the significant challenge of "imagining" the target state in open-world problems, suggesting that a world model capable of accurately providing such visual guidance requires further development.

**CogCoM**. We have also experimented with a potentially feasible baseline CogCoM (Qi et al., 2024). However, we find that it does not function well when tackling our challenges. Despite explicitly trained with a GROUNDING ability for counting objects, as shown in Tab. 7, directly prompting the model to output the counting result is more accurate than performing GROUNDING. When asked to complete the jigsaw puzzle by finding target location, it starts hallucinating by outputting unrelated text, as shown in Fig. 9. This is

potentially due to the model trained on different tasks and not generalizing to our challenged tasks. We decided to present the result in the appendix instead.

## C   Additional Qualitative Results

We present qualitative results on tasks that include counting, solving simple jigsaw puzzles, and placing objects in Fig. 10 - 14. These images more effectively illustrate the detailed visual reasoning processes and the changes within the imagination space. Please also see the supplementary video, which shows the reasoning process of our approach in the jigsaw puzzle solving task.

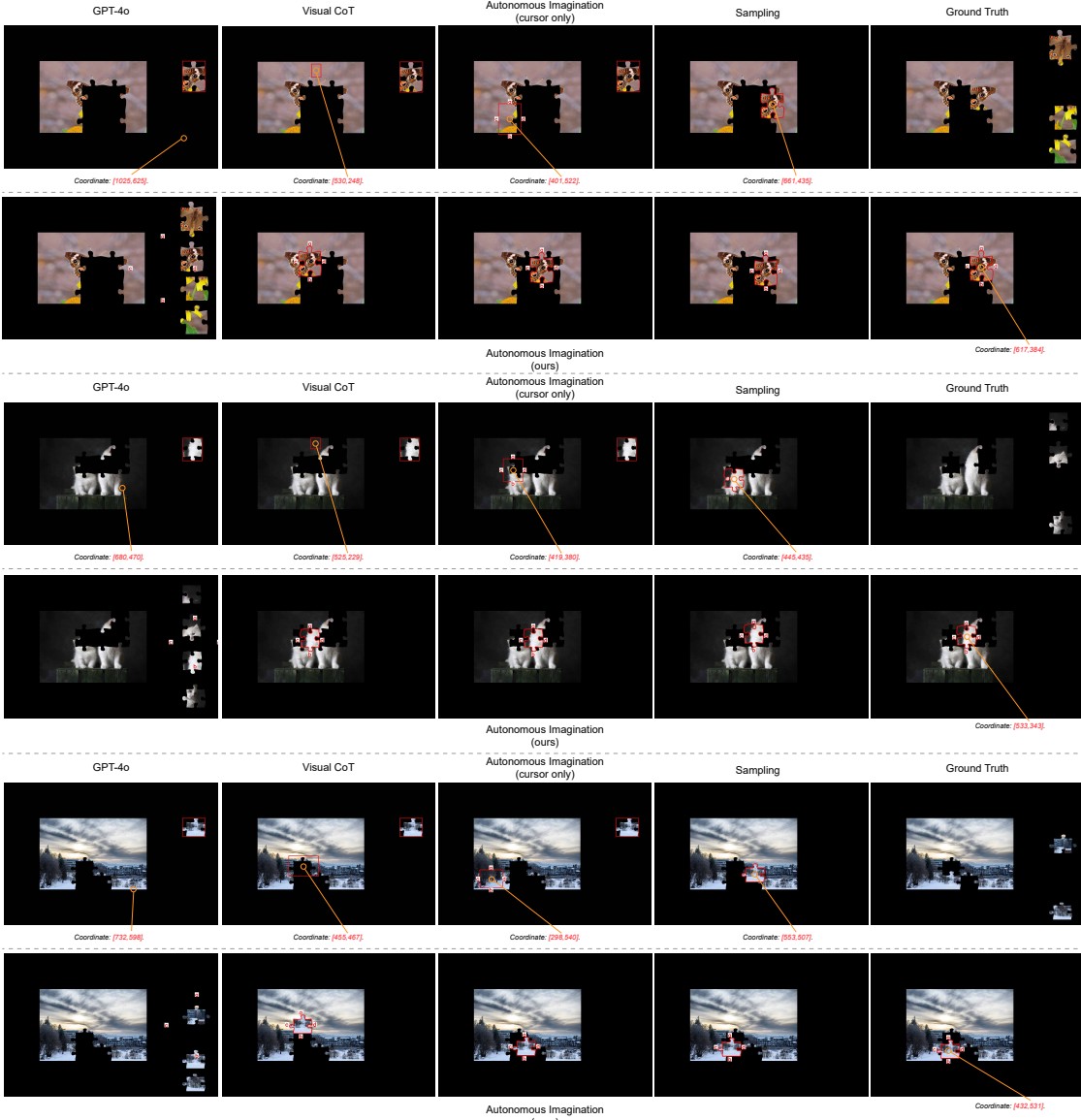

Figure 10: Additional qualitative results of simple puzzle solving. Please zoom in for a clearer view.

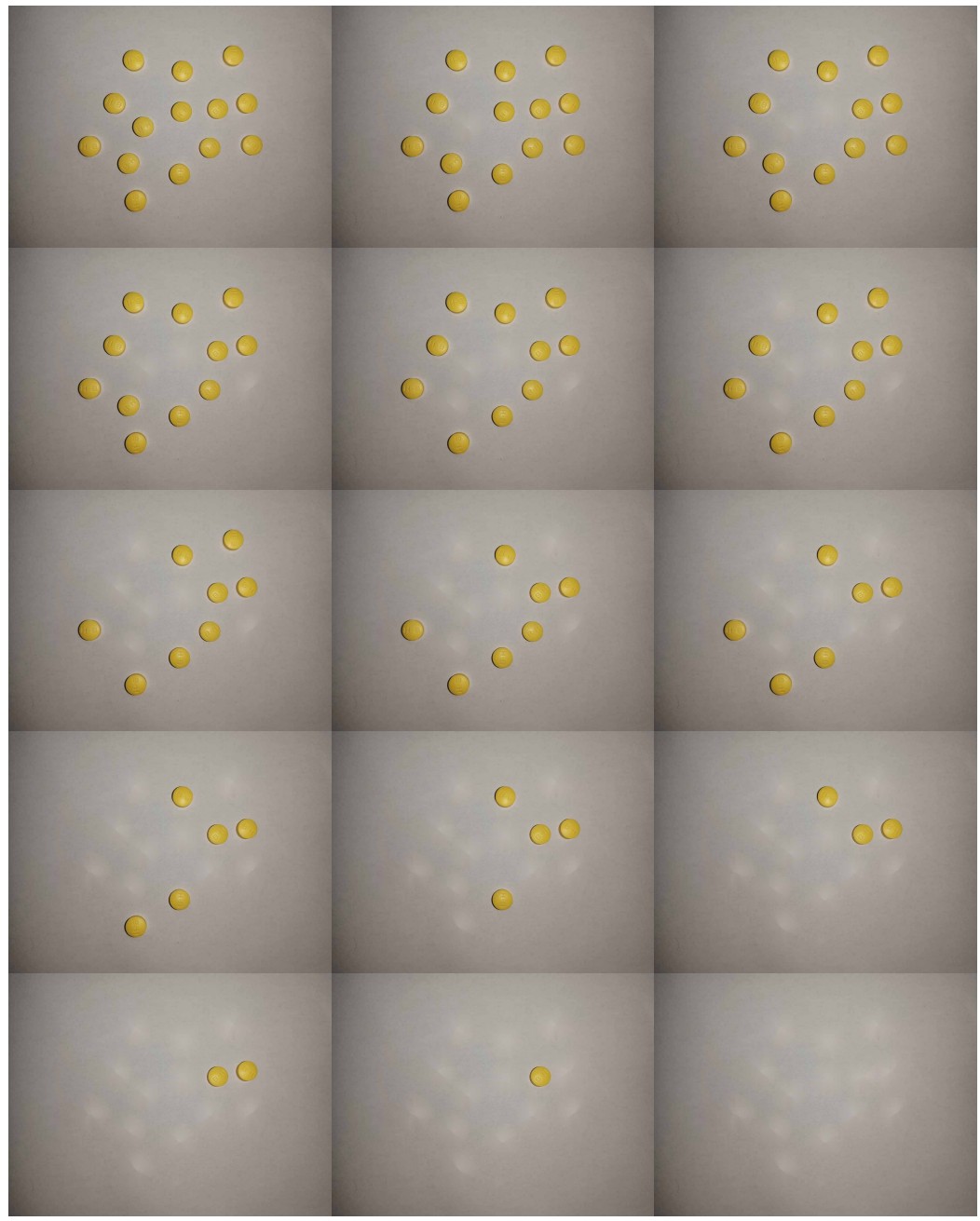

Figure 11: Qualitative demonstration of the counting process autonomously performed by our approach.

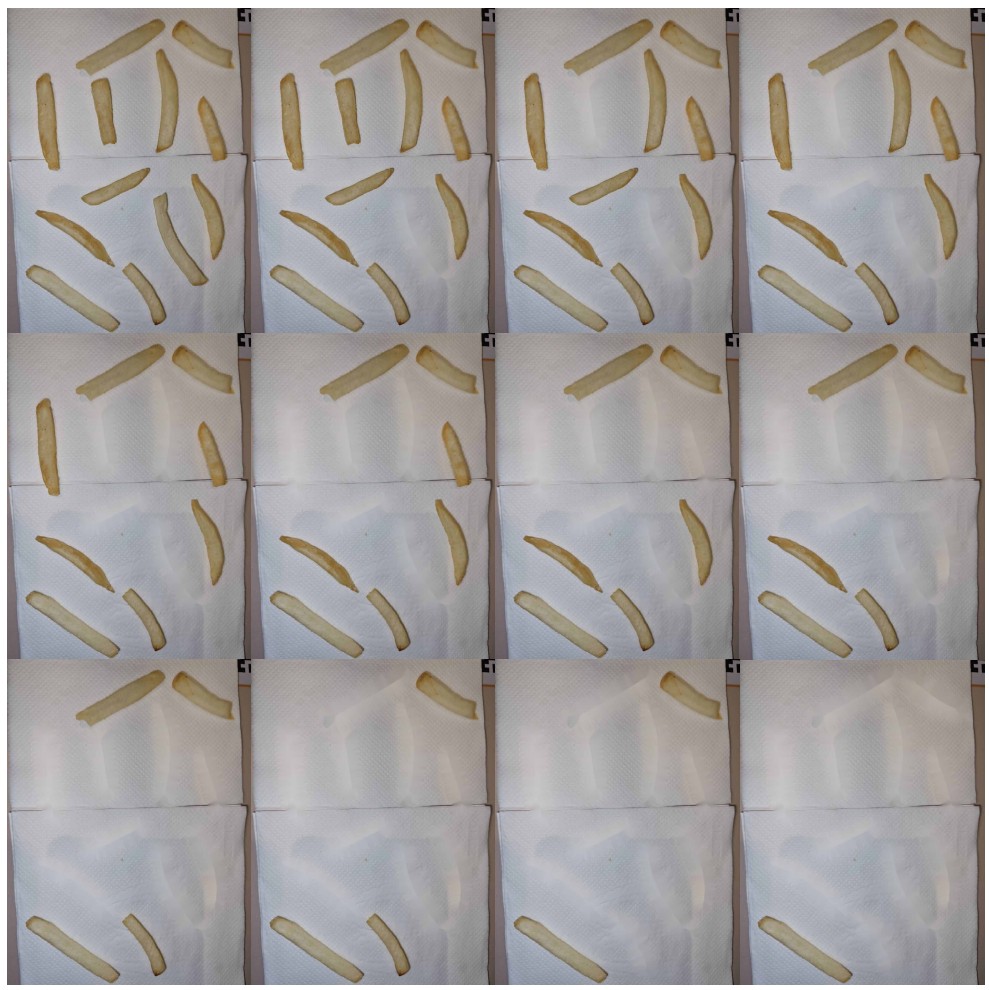

Figure 12: Qualitative demonstration of the counting process autonomously performed by our approach.

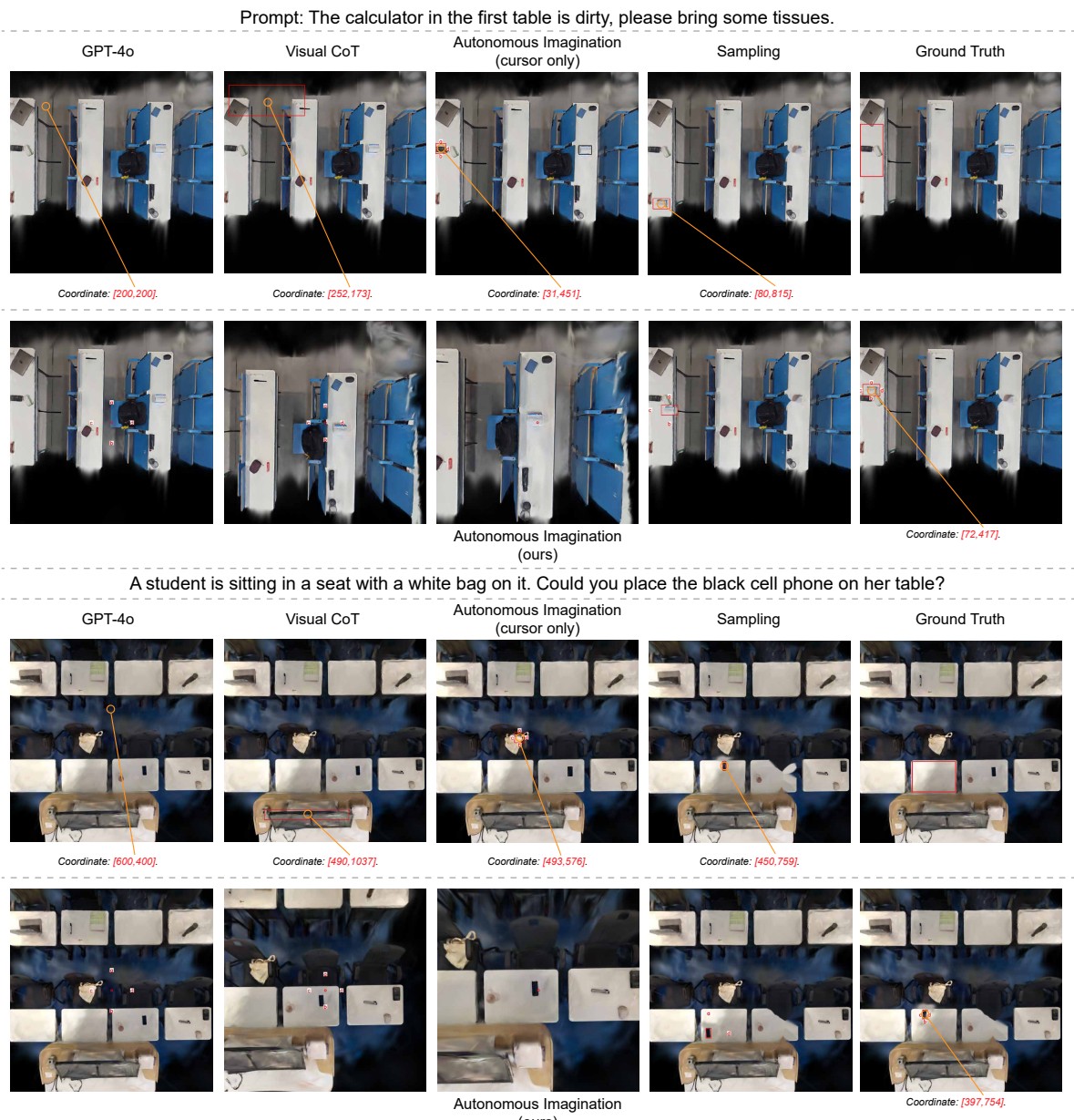

Figure 13: Additional qualitative results of object placement are provided. We also illustrate some steps in our method during the search for the target object and its placement. Please zoom in for a clearer view.

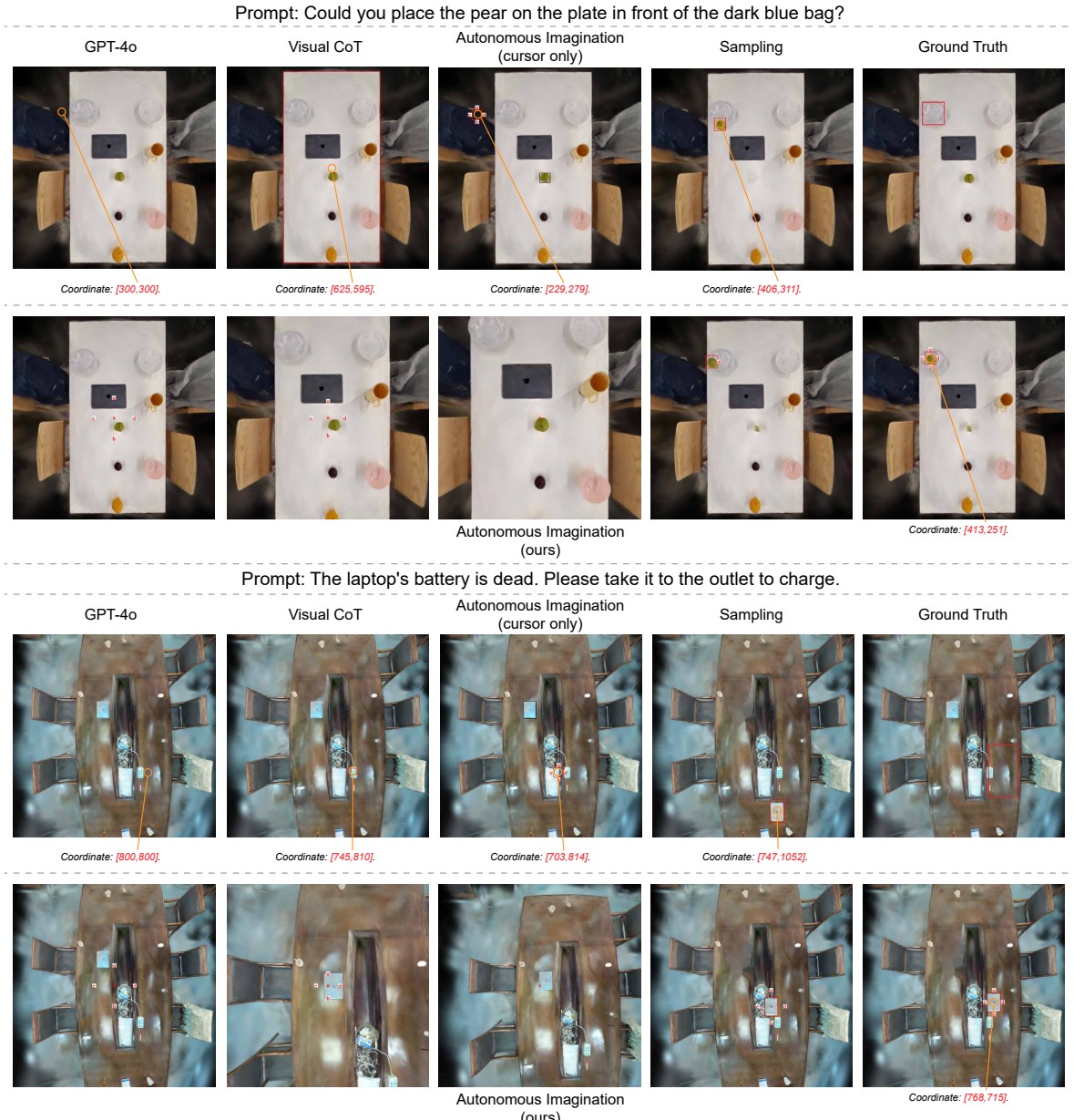

Figure 14: Additional qualitative results of object placement are provided. Please zoom in for a clearer view.

**General Reasoning Instructions for Counting**
You are an assistant which counts the number of specific objects in the given image and outputs it. Please recognize distinct objects one by one, describe each unique object including its color and location, and count during this process.

**General Reasoning Instructions for Jigsaw Puzzle Solving**
In this task, you will receive a picture of an unfinished puzzle. The image contains a nearly complete puzzle and several pieces that have not yet been placed into the whole.

You should think step by step as follows:
1. Which of these individual puzzle pieces fits best into the whole? Describe the contents and location of the puzzle piece.
2. Establish a two-dimensional rectangular coordinate system in the image, with the lower left corner as the origin, the right direction as the positive x-axis, and upward as the positive y-axis. The x-axis coordinate range of the image is 0~$IMAGE_WIDTH (pixels), and the y-axis coordinate range is 0~$IMAGE_HEIGHT (pixels).

Please output the coordinates of your selected piece in the image. You should output the answer in the last line in the format ['x_coordinate','y_coordinate'].

**General Reasoning Instructions for Object Placement**
You are an assistant that decides where to place an object based on user requirements.
First, make a plan by:
1. Listing the objects mentioned in the user requirement
2. Identifying which objects need to be moved or placed
3. Determining a reasonable order to move them

Finally, summarize the objects with numbers that need to be moved in the last line, using lowercase singular form. The output format should be [['Object', Object_number]].

A two-dimensional rectangular coordinate system is established in the image, with the lower left corner as the origin (right = positive x-axis, upward = positive y-axis). The x-axis ranges 0~$IMAGE_WIDTH (pixels), and y-axis ranges 0~$IMAGE_HEIGHT (pixels).

For each object:
1. Locate it by outputting its coordinates
2. Output the target location coordinates

You should format your final answer as ['x_coordinate','y_coordinate'].

**Prompts for Autonomous Imagination**
You can manipulate the scene using these operations:
1. Control a cursor to focus on objects
2. Transform focused objects or ignore them
3. Return to cursor control after operations

For counting: Ignore irrelevant objects while counting.
For object placement: Move the cursor to focus on the target object, then transform it to the proper location.
For jigsaw puzzles: Focus on the puzzle piece and transform it to its ideal position.

Figure 15: General reasoning instructions provided to models for visual reasoning, as well as the reasoning instruction given to utilize autonomous imagination. It is not difficult for an advanced MLLM to figure out this plan. As we aim to break the visual-to-textual conversion bottleneck, we simplify this by directly given the reasoning plan to the MLLM to avoid occasional failure caused by the MLLM choosing a bad reasoning strategy.

**Cursor Control**
You are now controlling the cursor. You should find the target object based on the plan you made. You will receive an image and the object to be located. There is a red circle in the image, surrounded by the four letters: 'a', 'b', 'c', and 'd'. You should choose the letter closest to the target object. If there are multiple target objects in the image, you should aim the red circle at the closest one. Ignore artifacts like shadows and floating spots. You should output the quoted letters in the last line, that is: ['a'], ['b'], ['c'], or ['d']. The distance will be reduced in each iteration until convergence, so you only need to choose the direction.

**Checking Focus Segmentation**
Your task is to check if the object shown in the image is the object you seek. The image should show a single target object against a black background. Areas of the image that do not belong to the object are set to black. If not, you should judge it as ['no'] and end your check. You should imagine what the object would look like from above and compare it to the image.

You should describe the object in the image, then output the check result in the last line in the format: ['yes'] or ['no'].

**After Focus**
You are now focusing on the object you desire. You can choose to perform a transform operation or ignore the operation based on your own plan. Output ['transform'] or ['ignore'] in the last line to perform the corresponding operation.

**Transform Control**
You are now moving the object to the destination. The object you are controlling is highlighted and surrounded by at most four letters: 'a', 'b', 'c', and 'd'. Ignore artifacts like shadows and floating spots. Choose the direction leading to the destination. The distance will be reduced in each iteration until convergence, so you only need to choose the direction. You should output the quoted letters in the last line, that is: ['a'], ['b'], ['c'], or ['d'].

**Cursor Control (Special Focus for Multi-Object Hallucination)**
You are now controlling the cursor. You should focus on the region based on the provided object ID. You should first move the cursor to the top-left corner of the object region. There is a red circle in the image, surrounded by the four letters: 'a', 'b', 'c', and 'd'. You should output the quoted letters in the last line, that is: ['a'], ['b'], ['c'], or ['d']. The distance will be reduced in each iteration until convergence, so you only need to choose the direction.

You have selected the top-left corner of the region. You should focus on the bottom-right corner of the region to be focused on. There is a red circle in the image, surrounded by the four letters: 'a', 'b', 'c', and 'd'. You should output the quoted letters in the last line, that is: ['a'], ['b'], ['c'], or ['d']. The distance will be reduced in each iteration until convergence, so you only need to choose the direction.

Figure 16: Detailed prompts instructing the MLLM about the operations it can utilize. This is given to the MLLM as a reminder when the MLLM enters corresponding state during the reasoning.

