# OpenReview forum: "Autonomous Imagination: Closed-Loop Decomposition of Visual-to-Textual Conversion in Visual Reasoning for Multimodal Large Language Models"
_TMLR — Accepted by TMLR_

### Review · Reviewer_KeX5 · 2025-07-06

**Summary Of Contributions:**

The paper addresses challenges faced by VLMs in performing intuitive visual reasoning tasks due to insufficient decomposition of the visual-to-textual conversion process. It convincingly argues that perceptual bottlenecks in VLMs cannot be overcome merely by scaling inference-time reasoning. To address this, the authors propose an approach named "autonomous imagination," allowing iterative modifications of visual inputs to generate intermediate visual states. This closed-loop visual modification effectively decomposes complex visual tasks into manageable substeps, enhancing performance without retraining. Experimental results show the effectiveness of this approach across challenging tasks, such as counting and jigsaw puzzle solving.

**Audience:**

Yes

**Claims And Evidence:**

No

**Requested Changes:**

Can you address the concerns raised in the above weakness section? My major concern would be the ad-hoc image operation design and possibly applying the framework in general benchmarks.

**Strengths And Weaknesses:**

I like the motivation of this paper, using some image operations during the inference time, for possible complicated visual tasks. The proposed framework is composed of reasonable components (while they are a bit ad-hoc IMO, please refer to the below). I appreciate the math illustration in Section 3.2.

Through experiments, they show that the proposed pipeline bring major improvements compared to GPT-4o and o1.


---

Regarding the weaknesses, I feel the design in this paper is somewhat ad-hoc. For example, as mentioned in Section 3.2, “we directly provide the MLLM with task-specific reasoning plans (e.g., ‘count by ignoring balls’) through system prompts.” Such approaches seem to benefit only very specific tasks rather than providing a generalizable design.

Additionally, the provided examples—“How many balls are there in this image?” and “The guest will sit on the right side of the table, near the yellow cup. Where should we serve the apple?”—are rather straightforward. These tasks do not appear to require the complicated autonomous imagination approach proposed.

So does the evaluation, the tested tasks "counting, "simple jigsaw puzzle", and "object placement" are very task-specific. I wonder if the authors provided separate, hand-tailored prompts for each individual task? Also, there are many other great open-source VLMs, such as QWen and UI-TARS. Can the author also try them? Finally, why only testing on openai-o1 not openai-o3?

---

> ### Author Response · Authors · 2025-07-27
> **responses to the review**
>
> **We sincerely thank the reviewer for the constructive and thoughtful suggestions.**
>
> - **Q1: The ad-hoc image operation design.**
> - A1:  We believe that the ignore, focus, and transform operators can be generally utilized in broader visual reasoning tasks. Furthermore, our approach can be implemented with a general set of system prompts as illustrated in Fig. 15 and Fig. 16. It can be seen that the prompts include brief descriptions of the tasks and reasoning plans (Fig. 15), and a set of imagination instructions shared for all tasks (Fig. 16). We have kept the design simple and clear enough to implement on different kinds of visual reasoning tasks. We have added Sec. 3.8 about the system prompts to make this clarified.  Considering the reasoning plan, it is not difficult for an advanced MLLM to figure out by themselves. As we focus on breaking the visual-to-textual conversion bottleneck, we simplify this by directly given the reasoning plan to the MLLM to avoid occasional failure caused by the MLLM choosing a bad reasoning strategy. Future research can indeed improve on this point, and we added the discussions in Sec. 5.
>
> - **Q2: Possibly applying the framework in general benchmarks.**
> - A2: We have added two existing benchmarks, CLEVR for counting, and Where2Place for object placement for further evaluating our approach. The results are discussed in Sec. 4.7 and Tab. 3.  Due to resource limitation, we are not able to conduct experiments on other general-purpose visual reasoning benchmarks, which may require specific model training. We have discussed this limitation in Sec. 5, and left it as an important future work.
>
> - **Q3: The provided examples are rather straightforward. These tasks do not appear to require the complicated autonomous imagination approach proposed.**
> - A3: As discussed in Sec. 1, the purpose of this paper is exactly to deal with visual reasoning tasks that seems to be straightforward for humans, while remains challenging for MLLMs. The key gap lies in the perceptual bottleneck in visual-to-textual conversion, which is the central technical challenge that our work tries to deal with. As discussed in Sec. 5, our current approach serves as an initial proof of concept that breaks the perceptual bottleneck. Future work can indeed train the model with native closed-loop reasoning and visual modification capabilities to approach and outperform human-level performance.
>
> - **Q4: If the authors provided separate, hand-tailored prompts for each individual task?**
> - A4: Please refer to A1.
>
> - **Q5: Try other open-source VLMs.**
> - A5: We have added discussions on both the limitations and potentialness to implement our approach on smaller open-source MLLMs in Sec. 4.7, and report the performance of Qwen2.5-VL and InternVL2.5 models under the object placement task in Tab. 4.
>
> - **Q6: Why only testing on openai-o1 not openai-o3?**
> - A6: Considering the costs of API calls, we choose GPT-4o as the major base model since it remains as one of the SOTA MLLMs with both strong visual perception and textual reasoning abilities. The openai-o1 baseline is introduced since its major advantage over GPT-4o lies in its specifically trained reasoning abilities under pure textual modality, while the visual perception ability remains relatively close to GPT-4o. This makes it a suitable choice to justify whether simply scaling text-time textual reasoning could solve the tasks. However, it is known that openai-o3 has more advanced perception and reasoning abilities in both visual and textual modalities over GPT-4o, which does not satisfy our purpose. This makes o1 a better choice than o3.

---

> > ### Comment · Reviewer_KeX5 · 2025-08-03
> > **Reviewer Response**
> >
> > I appreciate the author’s efforts, and the inclusion of Table 4 is very helpful.
> >
> > However, I believe the current justification for not using OpenAI-O3 is not fully convincing, as O3 currently does not employ any inpainting operations, which are the highlighted contributions proposed by this paper.
> >
> > Additionally, I noticed several related visual test-time scaling works were not properly cited in the manuscript. To avoid any potential conflict of interest, I prefer not to provide a detailed list here.
> >
> > Could the authors please address these concerns further? I can then summarize my review to the Action Editor, considering the new added experiments of CLEVR and Where2Place.

---

> > > ### Author Response · Authors · 2025-08-10
> > > **Thanks for the further suggestions**
> > >
> > > Dear Reviewer,
> > >
> > > Thanks for the further suggestions. To improve the paper accordingly, we have added an additional paragraph in Sec. 4.7, discussing "Concurrent work on multimodal CoT reasoning", including OpenAI-o3, visual test-time scaling works, and other recent concurrent studies on multimodal CoT reasoning. We have also reported the performance of o3 under our benchmarks in Tab. 6. If any concerns remain, please don't hesitate to let us know.
> > >
> > > We sincerely thank again for reviewer's efforts paid on our work.
> > >
> > > Authors

---

### Review · Reviewer_Ppp8 · 2025-07-11

**Summary Of Contributions:**

This paper aims to improve the visual reasoning ability of Multimodal Large Language Models (MLLMs). They argue that scaling textual reasoning alone is insufficient to solve visual tasks like counting. This paper proposes the autonomous imagination, which allows MLLMs to iteratively modify the visual inputs with pre-defined operations, including focus, transform, and ignore. The proposed approach is evaluated on four types of tasks: counting, jigsaw puzzles, object placement, and multi-object hallucination. The experiment results show that the proposed approach can enhance the performance of MLLMs on those tasks.

**Audience:**

Yes

**Claims And Evidence:**

Yes

**Requested Changes:**

1. Evaluate on more tasks, including some general vision-language tasks, existing benchmarks for counting, placing, etc.
2. Disentangle the impact of the lack of reasoning and the lack of in-domain training samples by including some fine-tuned models as baselines.

**Strengths And Weaknesses:**

Strength:
The proposed method is novel and intuitive. The experiments demonstrate that such autonomous imagination can significantly enhance the model's performance on some visual tasks.

Weaknesses:
1. **Generalizability of the operations.** The proposed three operations are quite customized to those evaluated tasks. Whether such types of imagination can help with other visual reasoning tasks is unclear. It may require manually designing new types of imagination for new tasks, thus limiting the scalability of the proposed methods. It would be valuable to include the evaluation on more general vision-language tasks to show that imagination at least won't hurt the performance.

2. **Assumption of single path reasoning.** The proposed method assumes a single-path reasoning trajectory, which means the model cannot revisit a previous state. In some challenging cases, models may make mistakes at a particular stage. Without a mechanism to revert to previous states, the model may never find a correct solution. Therefore, the proposed approach may excel at easy tasks that can be solved in one direction, but not those that require searching multiple routes.

3. **Disentangle the impact of lack of reasoning and lack of in-domain training samples.** From the results of counting in Table 1, Molmo achieves the top-1 performance without any reasoning process. The high performance of Molmo mainly comes from the high-quality counting training data. This reveals that the performance on some of those visual tasks can be enhanced by simply including in-domain training data. This may be applied to other tasks evaluated in this paper. For example, if we train the model to solve Jigsaw, do we really need inference time reasoning? Consider the higher cost of inference time scaling, maybe collecting training samples could be a more efficient way. To this end, adding experiments that demonstrate the limitations of scaling training data on certain tasks could help the community understand the necessity of inference time scaling for visual tasks.

4. **Evaluate on existing benchmarks.** Most of the tasks evaluated in the paper were designed by the authors, which makes the results less convincing. The counting benchmark is only a top-down view and is less likely to have cases like overlap, which could happen frequently in real photos. Including the results on existing benchmarks and directly comparing against numbers from prior works could make the results more solid. Some tasks to consider: Pixmo-count (Deitke et al), where2place (Wang et al).

---

> ### Author Response · Authors · 2025-07-27
> **responses to the review**
>
> **We sincerely thank the reviewer for the constructive and thoughtful suggestions.**
>
> - **Q1: Evaluate on more tasks.**
> - A1: We have added two existing benchmarks, CLEVR for counting, and Where2Place for object placement for further evaluating our approach. The results are discussed in Sec. 4.7 and Tab. 3.  Due to resource limitation, we are not able to conduct experiments on other general-purpose visual reasoning benchmarks, which may require specific model training. We have discussed this limitation in Sec. 5, and left it as an important future work.
>
> - **Q2: Disentangle the impact of the lack of reasoning and the lack of in-domain training samples by including some fine-tuned models as baselines.**
> - A2: We have included the discussions in Sec. 4.7, including the discussions on Molmo (fine-tuned on counting task) and RobotPoint (fine-tuned for the Where2Place task), and comparing the performance of them with our approach (See Tab. 1 and Tab. 5).  The results suggest that in-domain training is more suitable to address specific tasks, while improving test-time reasoning can be an easier way to improve general reasoning ability across different tasks.
>
> - **Q3: Generalizability of the operations.**
> - A3: We believe that the ignore, focus, and transform operators can be generally utilized in broader visual reasoning tasks. We utilize a unified set of operators for all tasks as shown in the system prompts in Fig. 16. We have added the introduction to the system prompts in Sec. 3.8 for better clarity.
>
> - **Q4: Assumption of single path reasoning.**
> - A4: We agree that optimization of the reasoning chain with advanced searching and planning strategies can improve the efficiency. Since we treat this work as an initial proof of concept, we left this as an important future work. We have added the discussions in Sec. 5.

---

### Review · Reviewer_tdza · 2025-07-18

**Summary Of Contributions:**

This paper introduces Autonomous Imagination, a plug-and-play framework that enables multimodal language models (MLLMs) to reason more effectively by iteratively editing the visual scene they’re analyzing. The method operates without any additional training and uses a set of simple yet powerful operators—focus, ignore, and transform, applied in an “imagination space” that supports both 2D and 3D representations. By decomposing visual reasoning into a series of step-by-step edits, the approach helps address the "perceptual bottleneck" that often limits MLLMs. The framework is tested on four visual reasoning tasks—counting, jigsaw puzzles, object placement, and hallucination mitigation—and shows consistent performance gains, especially in the counting task.

**Audience:**

Yes

**Broader Impact Concerns:**

There are no significant negative broader impact concerns raised in the paper. The work is largely foundational and aims to improve the reasoning capabilities and reliability of multimodal language models, which is a positive direction. Notably, its ability to reduce hallucinations in complex visual scenes could enhance safety and robustness in real-world applications. However, some softer concerns remain. The method incurs high computational costs due to repeated API calls to closed-source models like GPT-4o—raising issues of energy efficiency and environmental impact. Its dependence on proprietary models also limits accessibility and reproducibility, potentially disadvantaging researchers without access to these resources. Additionally, reliance on external tools such as SAM2 and OpenCV introduces the possibility of error propagation, which is not fully addressed. While not problematic in a foundational setting, these concerns merit discussion if the approach is deployed in broader or critical contexts.

**Claims And Evidence:**

Yes

**Requested Changes:**

To strengthen the paper, the authors should broaden the validation of the perceptual bottleneck hypothesis across all tasks. Introducing more granular levels of complexity in the jigsaw and hallucination tasks, or redesigning them to better reflect scalable visual difficulty, would help support the central claim more convincingly. Another important improvement would be to reduce the framework’s reliance on hand-crafted prompts. Allowing the MLLM to autonomously generate its reasoning plan from high-level instructions would greatly enhance the method’s generality and realism, aligning it more closely with the vision of autonomous multimodal agents.

In addition, the authors are encouraged to evaluate the approach on larger, publicly available benchmarks, particularly in the counting domain, to demonstrate its robustness and external validity. Running experiments on open-source MLLMs such as LLaVA-1.6 would also help show that the method is not tightly coupled to a single proprietary model and can generalize more broadly. Given the high computational cost, profiling the API calls in more detail and exploring optimizations like batching or pruning of operators could make the approach more practical. The paper would also benefit from increased clarity and accessibility: terms like G_marked should be clearly defined in the main text, and a brief primer on the 3DGS representation would help readers unfamiliar with 3D scene reconstruction. Finally, the authors should consider expanding the related work section to better position their contributions in light of concurrent research on visual scene editing and closed-loop planning.

**Strengths And Weaknesses:**

**##Strength##**

The paper presents a fresh perspective on visual reasoning by focusing on visual-to-text conversion as the core bottleneck and addressing it through closed-loop scene edits. The method is intuitive, modular, and works with existing MLLMs out-of-the-box—no retraining required. Experiments are thoughtfully designed and show strong improvements across multiple tasks. The ablation studies are convincing, and the paper is generally well-written with clear figures and detailed analysis. The use of generic operators and existing tools like SAM2 also makes the system relatively easy to reproduce and extend.

A key weakness lies in the paper’s heavy reliance on prompt engineering. Task-specific reasoning plans are hardcoded into prompts rather than autonomously generated, which limits the framework’s generality and scalability. This simplification, while acknowledged as a proof-of-concept, sidesteps the central challenge of enabling models to reason independently. Another concern is the limited validation of the core “perceptual bottleneck” hypothesis. While the counting experiments do convincingly show a performance drop as visual complexity increases, the other tasks—such as jigsaw puzzles, object placement, and hallucination mitigation—do not offer similarly strong or scalable evidence. These tasks lack sufficient variation in difficulty or do not directly measure how increasing complexity impacts model performance. Additionally, the evaluation is conducted on small, bespoke datasets, with only 122 images for counting and 17 scenes for object placement. This small scale makes it difficult to draw broad conclusions or assess statistical significance.


**##Weakness##**

The system also imposes a significant computational burden. The iterative reasoning process leads to a large number of model invocations—over 100 GPT-4o calls on average per image in some tasks—which may make it impractical for real-world or real-time applications. Furthermore, the method is currently tied to a proprietary model (GPT-4o), and its applicability to open-source models remains untested. This reliance raises concerns about accessibility, reproducibility, and long-term viability. The paper also includes some unclear or underdefined technical components. For example, key terms like G_marked are introduced in algorithm boxes but not fully explained in the main text, and heuristics such as the 10% voting rule in 3D segmentation are used without justification. Finally, while the related work section addresses visual prompting and chain-of-thought methods, it misses recent and relevant efforts in closed-loop VLM agents and visual planning approaches, such as ViperGPT or “Mind’s Eye,” which weakens the contextual framing of the contribution.

---

> ### Author Response · Authors · 2025-07-27
> **responses to the review**
>
> **We sincerely thank the reviewer for the constructive and thoughtful suggestions.**
>
> - **Q1: Introducing more granular levels of complexity in the tasks.**
> - A1: We indeed carefully considered to test the robustness of our approach under different levels of complexities of the tasks. Thanks for realizing our analysis on the counting task shown in Fig. 4. We also introduce two levels (4 and 6 pieces missing) of difficulties for the jigsaw puzzle solving task. While for object placement and multi-object hallucination tasks, properly measuring the task difficulty is itself a challenging problem. These tasks involve real-world scenes with complicated spatial and semantical relationships among the objects, which may not be easily captured by a single measurement of hardness. We treat this as an important future research problem, and added the discussions in Sec. 5.
>
> - **Q2:  Reduce the framework’s reliance on hand-crafted prompts.**
> - A2:  Our approach can be implemented with a general set of system prompts as illustrated in Fig. 15 and Fig. 16. It can be seen that the prompts include brief descriptions of the tasks and reasoning plans (Fig. 15), and a set of imagination instructions shared for all tasks (Fig. 16). We have kept the design simple and clear enough to implement on different kinds of visual reasoning tasks. We have added Sec. 3.8 about the system prompts to make this clarified. Considering the reasoning plan, it is not difficult for an advanced MLLM to figure out by themselves. As we focus on breaking the visual-to-textual conversion bottleneck, we simplify this by directly given the reasoning plan to the MLLM to avoid occasional failure caused by the MLLM choosing a bad reasoning strategy. Future research can indeed improve on this point, and we added the discussions in Sec. 5.
>
> - **Q3: Evaluate the approach on larger, publicly available benchmarks.**
> - A3: We have added two existing benchmarks, CLEVR for counting, and Where2Place for object placement for further evaluating our approach. The results are discussed in Sec. 4.7 and Tab. 3.
>
> - **Q3: Running experiments on open-source MLLMs.**
> - A3: We have added discussions on both the limitations and potentialness to implement our approach on smaller open-source MLLMs in Sec. 4.7, and report the performance of Qwen2.5-VL and InternVL2.5 models under the object placement task in Tab. 4.
>
> - **Q4: Profiling the API calls in more detail and exploring optimizations like batching or pruning of operators.**
> - A4: Our code will be publicly released to show all the implementation details, including the API calls. We agree that optimization of the reasoning chain with advanced searching and planning strategies can improve the efficiency. Since we treat this work as an initial proof of concept, we left this as an important future work. We have added the discussions in Sec. 5.
>
> - **Q5: Refining introductions to the methods and related work.**
> - A5: We have refined the corresponding parts in Sec. 3 and Sec. 4. Since our paper focuses on visual reasoning, the details of 3D Gaussian splatting reconstruction, rendering, and segmentation will be introduced in our released code for the interested readers.
>
> - **Q6: Discussions on potential broader impact.**
> - A6: We have added the discussion about the potential broader impact in the appendix.

---

### Author Response · Authors · 2025-07-27
**revised manuscript based on the reviews**

We would like to thank all the reviewers for the constructive and insightful suggestions. We have revised the manuscript accordingly. The details are discussed in the responses to the reviewers.

---

### Decision · Action_Editor_mPNG · 2025-08-27

**Recommendation:** Accept as is

**Audience:**

Yes

**Audience Explanation:**

The notion of visual reasoning in the visual image space is interesting and novel. The authors have shown early success that the basic image editing can improve counting task. This could inspire future research to further push the direction for more comprehensive visual reasoning and improved performance on a wide range of tasks.

**Claims And Evidence:**

Yes

**Claims Explanation:**

- *The authors introduce a novel visual reasoning method, by editing in image space*.  This is supported by the proposed method "autonomous imagination", which empowers Multimodal Large Language Models (MLLMs) to iteratively transform visual inputs—such as isolating objects or rearranging puzzle pieces—into a sequence of intermediate visual states. This process effectively decomposes the visual-to-textual conversion task into a series of closed-loop visual modification steps.

- *Improved performance is shown*. Without any additional training, MLLMs can successfully tackle tasks that were previously beyond their perceptual capabilities. This finding underscores the potential of closed-loop visual modification as a powerful strategy for breaking down complex visual reasoning tasks into manageable subproblems. The improvement is shown especially in the counting task.

---

> ### Author Response · Authors · 2025-09-17
> **camera ready version**
>
> Dear Action Editor and Reviewers,
>
> We have submitted the camera ready version of the paper. We are grateful for your time, thoughtful feedback, and constructive guidance, which have greatly contributed to improving the quality of our submission.
>
> Best regards,
>
> Authors